# Improving the Accuracy of Saffron Adulteration Classification and Quantification through Data Fusion of Thin-Layer Chromatography Imaging and Raman Spectral Analysis

**DOI:** 10.3390/foods12122322

**Published:** 2023-06-09

**Authors:** Haochen Dai, Qixiang Gao, Jiakai Lu, Lili He

**Affiliations:** 1Chenoweth Laboratory, Department of Food Science, University of Massachusetts Amherst, 102 Holdsworth Way, Amherst, MA 01003, USA; 2Department of Chemistry, University of Massachusetts, Amherst, MA 01002, USA

**Keywords:** Raman spectroscopy, saffron, adulteration, thin layer chromatography, data fusion, imaging data, classification, quantification

## Abstract

Agricultural crops of high value are frequently targeted by economic adulteration across the world. Saffron powder, being one of the most expensive spices and colorants on the market, is particularly vulnerable to adulteration with extraneous plant materials or synthetic colorants. However, the current international standard method has several drawbacks, such as being vulnerable to yellow artificial colorant adulteration and requiring tedious laboratory measuring procedures. To address these challenges, we previously developed a portable and versatile method for determining saffron quality using a thin-layer chromatography technique coupled with Raman spectroscopy (TLC-Raman). In this study, our aim was to improve the accuracy of the classification and quantification of adulterants in saffron by utilizing mid-level data fusion of TLC imaging and Raman spectral data. In summary, the featured imaging data and featured Raman data were concatenated into one data matrix. The classification and quantification results of saffron adulterants were compared between the fused data and the analysis based on each individual dataset. The best classification result was obtained from the partial least squares—discriminant analysis (PLS-DA) model developed using the mid-level fusion dataset, which accurately determined saffron with artificial adulterants (red 40 or yellow 5 at 2–10%, *w*/*w*) and natural plant adulterants (safflower and turmeric at 20–100%, *w*/*w*) with an overall accuracy of 99.52% and 99.20% in the training and validation group, respectively. Regarding quantification analysis, the PLS models built with the fused data block demonstrated improved quantification performance in terms of R^2^ and root-mean-square errors for most of the PLS models. In conclusion, the present study highlighted the significant potential of fusing TLC imaging data and Raman spectral data to improve saffron classification and quantification accuracy via the mid-level data fusion, which will facilitate rapid and accurate decision-making on site.

## 1. Introduction

Saffron, the stigma of *Crocus sativus* L., which is also called “the red gold”, is one of the most expensive agricultural products on the market. For centuries, saffron has been utilized as a medicinal herb, spice, and colorant [1,2,3,4,5]. Throughout history, saffron has been vulnerable to economic adulteration, which involves the mixing of low-quality spices with saffron, the addition of plant materials, and the use of natural or artificial colorants to imitate the color of saffron [6].

Color strength is one of the key attributes used to describe saffron quality. The current standard method for saffron color strength analysis is protocol ISO 3632-2, issued by the International Organization for Standardization [7,8]. This method uses UV-vis spectroscopy and high-performance liquid chromatography (HPLC) to determine the color strength and crocin content, which is the active ingredient responsible for saffron’s color expression. Despite its widespread use, this method still has several limitations. For instance, some studies have reported that UV-vis spectroscopy can only detect saffron adulterants up to 20% *w*/*w* and is unable to distinguish between artificial yellow adulterants that display similar absorbance values to pure saffron [9,10]. Other well-adopted methods, such as HPLC and gas chromatography (GC), still have limitations such as long sample preparation, long test duration, high costs, and the need for specialized laboratory settings and trained personnel.

Previously, we developed a rapid and field-deployable method based on thin-layer chromatography (TLC) and Raman spectroscopy to determine saffron quality as well as saffron adulteration. By utilizing TLC as a separation substrate, the optical TLC pattern of pure saffron and adulterated saffron specimens could be easily captured via a camera and converted to digital imaging data under ambient light and 365 nm UV light [10]. Raman spectroscopy, in the meantime, provides molecular information on pure or adulterated saffron specimens from the TLC chip [11]. The established TLC-Raman method has demonstrated its capability to determine saffron grades and identify common adulterations. However, upon further execution of this method for adulterant quantification, we identified several notable drawbacks. In general, Raman signals provide information for quantifying the purity of saffron, while TLC patterns are used primarily for identifying saffron adulterations. Unfortunately, the lack of communication between these two data blocks limits the full potential of the method from being exploited. For instance, the imaging method failed to accurately determine adulteration levels at high yellow 5 concentrations (6–10% *w*/*w*) due to color saturation on TLC chips. In this case, the decrease in crocin concentration caused by yellow 5 adulteration can still be detected by the Raman spectrometer. Nonetheless, without communication between these data blocks, the final decision-making process relies on separate data analyses of the TLC pattern and Raman spectroscopy.

Data fusion, the analysis of concatenating several datasets into a single fused data block, has shown great potential to improve existing performance in spectroscopic analysis [12,13,14]. The integration of multiple datasets through data fusion enables interactivity and mutual information among each data block, resulting in a reduction of spurious sources of variability and diminished prediction errors when compared to analyses based on individual datasets. The concatenation of data can be carried out and categorized at three different levels: data level (low-level fusion), feature data level (mid-level fusion), and decision level (high-level fusion). With spectral data, data level and feature data level fusions are more suitable to be executed. In low-level data fusion, data from all measurement sources are simply combined into a single matrix after some data pretreatments for each individual data block. Mid-level data fusion involves the extraction of relevant features from each data block separately, which are then concatenated to form a single data block. Features can be defined as either relevant original variables or latent variables that are extracted through multivariate analysis models [12,13,14,15].

Extensive data fusion research has been conducted in the fields of food quality analysis and food fraud analysis. Most of these studies have yielded positive outcomes when employing fused datasets [16]. The integration of similar datasets, such as mass spectrometry, UV-vis, near-infrared (NIR), mid-infrared (MIR), or Raman spectral data, has been a primary focus in classifying or quantifying food quality attributes, such as the quality of soybeans [17], characterizing olive oil and essential oils [18,19], and determining the geographical origins of wines and saffron [20,21].

The food industry has efficiently utilized imaging data fusion techniques, such as E-eye or computer vision, for post-harvest quality analysis of fruits, providing advantages in terms of speed and cost-effectiveness [16]. Moreover, imaging analysis has garnered significant attention in various food categories. This includes assessing the sensory scores of fish fillets [22], evaluating apple fruit firmness [23], and detecting bruises in strawberries and blueberries [24,25]. Additionally, researchers have explored the fusion of imaging data with spectral data, which has proven successful in monitoring the fermentation quality of black tea and classifying green tea [26,27]. The aim of this paper was to evaluate the effectiveness of data fusion methodologies to improve the classification and quantification accuracy of adulterated saffron using Raman and TLC imaging data. In this study, a mid-level data fusion strategy was evaluated for classification and quantification of saffron quality, respectively. In brief, saffron specimens were adulterated with artificial colorants or extraneous natural plant materials at different levels, from 2% to 100%. Both adulterated and pure saffron specimens were prepared into sample solutions before each sample droplet was developed on TLC chips. Then raw imaging data and Raman spectra were collected on the TLC chip, respectively. Subsequently, featured imaging data were extracted by taking L*a*b* values at characteristic locations in the image (that were, edge and midplane), while featured Raman data were generated through variable influence on projection analysis (VIP) of the crocin characteristic Raman peaks. At last, the fused data matrix was completed by concatenating the featured imaging data and the featured Raman data. The fused data were used for multivariate classification and regression via partial least squares discriminant analysis (PLS-DA) and partial least squares analysis (PLS), respectively. The classification and quantification results of saffron adulterants from the fused data were also compared against the analysis results based on each single dataset.

## 2. Materials and Methods

### 2.1. Chemicals and Reagents

All natural specimens, i.e., saffron, turmeric powders, and safflower, were purchased from online sources. Each plant material was acquired from three different suppliers with a similar price range. Artificial colorants (Allura red and tartrazine) were purchased from two suppliers, which were IFC Solutions (Linden, NJ, USA) and Sigma Aldrich (Merk KGaA, Darmstadt, Germany). TLC aluminum plates (Silicagel 60W F254S) were purchased from EMD Millipore Corporation (Billerica, MA, USA).

### 2.2. Sample Preparation

All plant materials were pulverized using a Newtry high-speed food mill (Guangzhou, China) set at high speed for three minutes. Sample particle sizes were standardized by sieving through a 500 µm mesh sieve after pulverization of raw plant material. The resulting powders were then transferred into glass vials and stored in a light-shielded desiccator. Natural adulterants (safflower and turmeric) were predominantly utilized to manipulate the weight of saffron samples. To calibrate the analysis for the presence of these adulterants, samples were spiked with varying weight percentages of pure saffron (0%, 20%, 40%, 60%, 80%, and 100%). Lower spike levels for natural adulterants were considered, but it was deemed impractical to use trace levels of these plant materials for the purpose of adulteration in normal practices. To achieve the desired appearance and color intensity, artificial adulterants (tartrazine yellow and Allura red) were typically added to saffron in lower amounts compared to natural adulterants. Hence, we used weight percentages of 0%, 2%, 4%, 6%, 8%, and 10% to calibrate the level of artificial adulterants mixed with pure saffron. For all samples, 50 mg of pure or spiked powdered samples were dissolved in 50 mL of deionized water to prepare raw sample solutions. The untreated solutions were subsequently subjected to filtration using a 0.45 µm PES filter (GE Lifesciences, Marlborough, MA, USA). Subsequently, 2 µL of pure saffron or saffron-spiked solutions were deposited onto TLC plates using a pipette. Three droplets of the same sample solution were applied to the same location on the TLC plate, ensuring that the preceding droplet had completely dried before adding the next one.

### 2.3. Data Acquisition

#### 2.3.1. Raw and Featured Raman Spectral Data

Raw Raman spectral data were collected at the center of each TLC pattern by a portable Raman system (TSI Incorporated, Shoreview, MN, USA), which was equipped with a 780 nm laser source. Each sample was measured in triplicate with five data collection spots for each replicate with an acquisition time of 10 s. The measurements were carried out at max laser power (500 mW) with a spectral range of 100 to 2200 cm^−1^. Thus, 75 spectra were collected for each type of adulteration, and 15 spectra were collected for pure saffron. A total of 315 spectra were collected for data analysis in this study. The Raman spectrum of each adulterated sample at different adulteration concentrations can be found in Appendix A. Figure 1 shows the adulterated saffron Raman spectra, which were collected on TLC silica gel substrates. Featured Raman data (1000~1050 cm^−1^, 1130~1240 cm^−1^, 1270~1300 cm^−1^, and 1500~1580 cm^−1^) were extracted by reviewing the main compounds responsible for saffron color expression. Feature selection was also referred to as the result of the variable influence on projection (VIP, VIP value > 1) from the SIMCA software (Malmö, Sweden, version 14.1), which summarizes the importance of each Raman peak in the PLS-DA model.

#### 2.3.2. Raw and Featured Imaging Data

The collection of raw imaging data and the process to generate the featured imaging data from TLC chips could be found in our previous work with some modifications [10]. In brief, two separate images were taken under ambient and UV (365 nm) light conditions. The ambient and UV images were then combined into a single image, as depicted in Figure 2A. To collect imaging data, Adobe Photoshop CS6 (64-bit) was utilized to split RGB images into the red, green, and blue channels (Figure 2C). A color picker tool (32 × 32 pixels square) was used to collect 20 data points from each TLC pattern per channel, with 10 points taken at ½ diameter distance from the center and 10 points taken from the TLC ring. This was performed for both the ambient light image and the UV light image under the green and blue channels, respectively (Figure 2C). The red channel data were excluded due to insufficient pattern information on the image (Appendix A).

The featured imaging data can be expressed as shown below:
$$X_{Ambient}=\left[X_{ambient\;green\;channel\;data},X_{ambient\;blue\;channel\;data}\right]$$
$$X_{UV}=\left[X_{UV\;green\;channel\;data},X_{UV\;blue\;channel\;data}\right]$$
$$X_{Featured\;Imaging}=\left[X_{Ambient},X_{UV}\right]$$
where the maximum, minimum, and average lightness values at the blue and green channels (Figure 2C) were collected at each sample collection point. Each adulteration level was measured in triplicate, resulting in a total of 7560 featured imaging data collected (7200 for adulterated saffron and 360 for pure saffron).

### 2.4. Data Fusion

Figure 3 illustrates the fusion strategy, where featured Raman data and featured imaging data were measured and fused into one data block in the mid-level fusion model. Normalization was applied to both data blocks prior to the fusion according to the equation below [28].
$$x_{i}^{*}=\frac{x_{i}-\min\left(X\right)}{\max\left(X\right)-\min\left(X\right)}$$

The fused data can be expressed via the equation below:
$$X_{Fused}=\left[X_{Featured\;Raman},X_{Featured\;Imaging}\right]$$
where *X_Fused_* stands for the fused data block, *X_Featured Raman_* and *X_Featured Imaging_* represent the single data blocks of featured Raman and featured imaging data, respectively. Next, *X_Fused_*, *X_Featured Raman_* and *X_Imaging_* data blocks were loaded in PLS-DA and PLS models for multivariate qualitative and quantitative analyses.

### 2.5. Partial Least Squares Discriminant Analysis

Partial least squares discriminant analysis (PLS-DA) was used to conduct classification analysis in SIMCA (14.1) software. The entire classification model was classified by adulterant type using Pareto scaling. This scaling method was chosen due to its ability to retain the proximity of the scaled measurements to the original data, minimizing disturbances to the raw data, minimizing disturbances to the feature data, and ensuring more reliable feature peaks that are less vulnerable to noise [29]. The significant compounds were optimized via cross-validations. In the meantime, a total of 210 files from each dataset were selected as a training group, which included 50 spectra for each adulterant and 10 spectra for pure saffron. As a result, 3D scattered point graphs (Figure 4) and a misclassification table (Table 1) were generated. In addition, the performance of the data fusion strategy was evaluated by the remaining 105 spectra working as an external validation group, which included 25 spectra for each type of spiked saffron and 5 spectra of pure saffron (Table 2).

### 2.6. Partial Least Square (PLS)—Regression of Quantification

The PLS regression model was used to determine saffron adulteration levels. Raw data, featured data, and fused data from the classification study were directly used in the quantification analysis. Spectral data and imaging data were set up as variable X, whereas the adulteration level was set up as variable Y. The result was presented as a predicted value versus an observed value, as shown in Figure 5. The performance of each PLS plot can be partially expressed by the maximum of R^2^ values (goodness of fit) or the minimum of the root-mean-square error of prediction (RMSEP) values and the root-mean-square of cross-validation (RMSECV) values.

## 3. Results and Discussion

### 3.1. The Effect of Mid-Level Fusion on Classification Performance

As is shown in Figure 4A, the featured imaging data itself showed a clear cluster separation for safflower, turmeric, and yellow 5 adulterated samples. However, heavy cluster overlapping was observed between turmeric and red 40, yellow 5, and pure saffron clusters. These results were also reflected in the misclassification table. In Table 1A, 20 out of 50 turmeric samples were misclassified as red 40. In addition, all pure saffron samples were misclassified as yellow 5 adulterated samples. This was due to the low discrimination between these samples on TLC chips, which was also reported in our previous study [10].

In Figure 4B, the featured Raman data generated more scattered clusters for samples with natural adulterants. On the other hand, samples with artificial adulterants overlapped with pure saffron and thus led to misclassification in Table 1B. This was due to the low spike level (2–10% *w*/*w*) for artificially adulterated specimens as compared to the one for natural adulterants (20–100% *w*/*w*). The reason for spiking lower concentrations of artificial adulterants was their superior color intensity compared to natural adulterants. Therefore, less concentration was required to achieve the same color appearance as the natural adulterants. However, this posed analytical challenges when detecting these artificial adulterants. The result of mid-level data fusion on classification accuracy can be seen in Figure 4C and Table 1C. Both the PLS-DA plot and the misclassification table indicated significant improvements in cluster separation and correction rate compared to the results from each individual data block. In the meantime, the PLS-DA plot with mid-level fused data exhibited tighter clusters, yet each class was distributed with better separation, especially for red 40 and yellow 5 adulterated specimens (Figure 4C). Each individual data block provided a complementary piece of information that helped in the classification of spiked saffron samples. For instance, red 40 and turmeric-spiked saffron samples that could not be clearly differentiated using the imaging data (Figure 4A and Table 1A) were separated in the PLS-DA plot when featured Raman data were fused. Similarly, the introduction of the imaging data also helped enhance the classification capabilities of the featured Raman data to discriminate specimens with red 40 and yellow 5 adulterants (Figure 4B and Table 1B). The collaborative effect between these two data blocks in the fused matrix achieved a satisfying accuracy of 99.52% (Table 1C). These findings were in line with most reported results, indicating that data fusion yielded clear improvements in classification accuracies compared to individual analytical methods. For instance, enhanced classification results in identifying hazelnut paste adulteration were reported by combining FT-Raman and NIR spectroscopy as the fused dataset [30]. Likewise, the classification of the geographical location of a medicinal plant (*Gentiana rigescens*) demonstrated improved performance through the fusion of UV-vis and infrared spectroscopy data [31].

### 3.2. Mid-Level Fusion Model Validation for Adulteration Classification

The performance of the PLS-DA model was validated using external validation samples (*n* = 105), and the results are shown in Table 2. It is clear to see that the validation group had an excellent classification result among all adulterated specimens. Most samples with safflower, turmeric, and red 40 adulterants were all correctly identified, with adulteration levels ranging from 2–10% *w*/*w* for artificial adulterants and 20–100% for natural adulterants. The model achieved a correct rate of 99.2%, in which no adulterated specimens were classified as pure saffron in the validation group, giving it a 100% accuracy rate on adulterated sample determination.

### 3.3. The Effect of Data Fusion on PLS Model Quantification Performances

Figure 5 shows the PLS plots that predict spike levels in different adulterated saffron samples. Three data blocks, namely, *X_Featured Imaging_*, *X_Featured Raman_*, and *X_Fused_*, were used to build each PLS plot, respectively. The R^2^ and RMSECV values in Table 3 were used to describe the goodness of fit of the model and the model’s ability to predict unknown samples. The RMSEP values indicate the goodness of the prediction of the model using external validation samples (*n* = 25). However, it is worth mentioning that the RMSECV and RMSEP values are positively correlated to the scale of the data. In this case, samples with natural adulterants (20–100% *w*/*w*) were expected to have larger RMSECV and RMSEP values than samples with artificial adulterants (2–10% *w*/*w*). Thus, artificial adulterants (red 40 and yellow 5) and natural adulterants (safflower and turmeric) should be compared separately due to the different adulteration levels between these two types of adulterants.

Separately, the PLS plots based on the featured imaging and Raman data in Figure 5C(a,b),D(a,b) showed poor prediction performances in the cases of red 40 and yellow 5 adulterated samples. As previously mentioned, low spiking levels were the main reason behind this phenomenon. In addition, sample droplets suffered from diffusion problems in every direction on TLC substrates. Thus, the actual concentrations of the target adulterants on the chips were usually lower in the test spots. This issue can potentially be solved by concentrating the sample solution or dropping multiple sample droplets at the same spot. However, this approach may bring new detection challenges, such as stronger interference from crocin.

When the fused data was used, higher R^2^ and lower prediction errors were achieved in most cases (Table 3). The best improvement was observed in Figure 5D(c), where the fused data block of yellow 5 adulterated specimens produced a significant improvement in the fittings of the plot. This result indicated the existing collaborative effects when the imaging and featured Raman data were combined. These effects, although not significant for most adulterated samples, still helped improve the quantification capabilities of the model.

When using actual validation samples, a similar trend could be seen with the RMSEP values. That is, the model had better performance using the fused data block for most adulterated samples, showing lower errors of prediction.

Similar studies also reported that the integration of independent datasets, such as the fusion of electronic nose and electronic tongue data, resulted in improved regression models for predicting juice quality parameters, including pH, titratable acidity, vitamin C, total soluble solids, phenolic content, and color indices of red wine. In addition, these studies consistently found higher R^2^ values in both the calibration and validation sets for the combined dataset compared to each individual measurement in most cases [32,33].

Nevertheless, it is worth noting that the improvement of the fusion strategy on prediction accuracies varied across different parameters or attributes. While certain specific variables demonstrated significant enhancements in prediction accuracies as a result of the fusion approach, the majority of variables experienced only modest improvements. Moreover, upon analyzing the quantification plots, it was observed that the fused dataset closely resembled the highest-performing individual dataset, indicating the preservation of key characteristics and similarities between the fused data and the most optimal standalone dataset [32,33].

Despite the overall improved quantification performance achieved with the fused data, there were some exceptions that brought some concerns about the effectiveness of the model. For example, the reduction in RMSEP and RMSECV values was not consistent when the fused data were used in PLS models. Samples spiked with safflower and red 40 showed slightly higher RMSEP and RMSECV values in the fused data, respectively. It was reasonable to believe that sample variation might have caused this issue. However, it still exposed one weakness of utilizing data fusion in quantification analysis. That is, the concatenation of different data blocks might cause deleterious effects when bad data blocks or outliers were introduced. This problem might not be obvious in sample classification since it is in favor of sample variation. However, it was very sensitive to quantification analysis. This inconsistent phenomenon was discussed in similar studies, where the presence of redundant information or the absence of feature information could impair the effectiveness of the fused data block [34,35]. This could be attributed to inappropriate feature extraction methods, which may result in datasets that are data-rich but information-poor or lead to the loss of important feature information [16,36]. In the present study, the selection of featured imaging data, such as pattern location and data acquisition numbers for each collection area, was determined manually. Consequently, it would be challenging to evaluate which manually defined imaging texture or pattern was the most sensitive to different adulteration concentrations. Furthermore, in this study, all variables in the fused data were uniformly weighted, assuming equal importance for classification or quantification predictions. Although this approach was proven suitable for classification work, a variable-wise weighted fusion approach may be more appropriate for quantification tasks [36].

## 4. Conclusions

The present study examined the effectiveness of mid-level data fusion strategies on the improvement of saffron adulteration classification and quantification accuracies using TLC imaging data and Raman spectral data. Our results indicated that mid-level data fusion had excellent classification performance. Under this setting, all spiked samples were identified and distinguished from pure saffron samples. In the meantime, the validation result exhibited great capabilities to distinguish each adulterant with excellent accuracy (safflower, 100%; turmeric, 100%; red 40, 96%; yellow 5, 100%).

In the meantime, quantification accuracies of both artificially and naturally adulterated samples showed better performances using the fused data block, achieving higher R^2^ values with lower errors. However, upon analyzing the results from external validation samples, it was found that the improvements in quantification accuracies were not markedly significant in comparison to the advancements observed in the classification results.

Further work needs to be performed to establish better protocols for using imaging data in quantitative analysis. This could involve assessing various strategies for imaging feature extraction to further improve the accuracy of quantification. Other data blocks can also be investigated and introduced to finalize both classification and quantification models. Furthermore, the established result can be further concatenated to produce a high-level fused result, which consists of both classification and quantification results. First, classification results are obtained from mid-level data fusion using the PLS-DA model. Then, corresponding quantification algorithms will be chosen based on the classification result. Depending on the classification results, the quantification results could be expressed either as saffron grade or adulterant level. Eventually, the final developed algorithm would be able to automatically determine saffron authenticity, adulterant identification, spike level, and pure saffron grade at the same time to facilitate fast decision-making on site.

## Figures and Tables

**Figure 1 foods-12-02322-f001:**
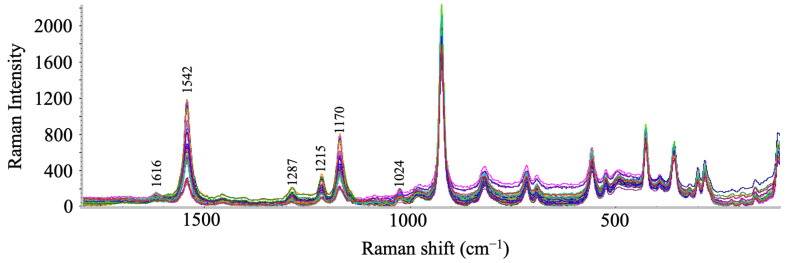
Raman spectra of adulterated saffron samples collected on TLC substrates.

**Figure 2 foods-12-02322-f002:**
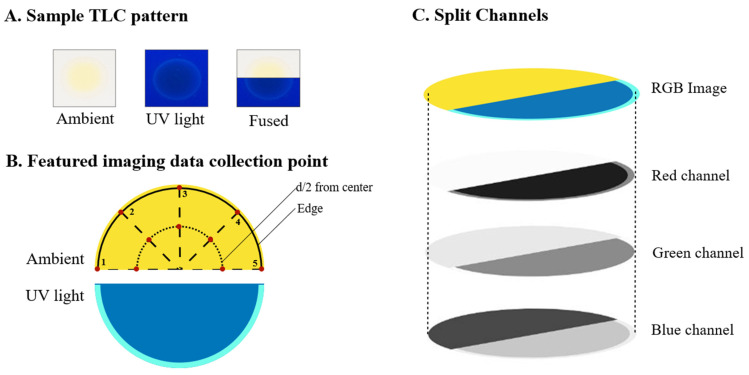
(**A**) TLC pattern of adulterated saffron specimen under ambient and UV light; (**B**) featured imaging data collection point; (**C**) channel split in imaging processing software (Adobe Photoshop).

**Figure 3 foods-12-02322-f003:**
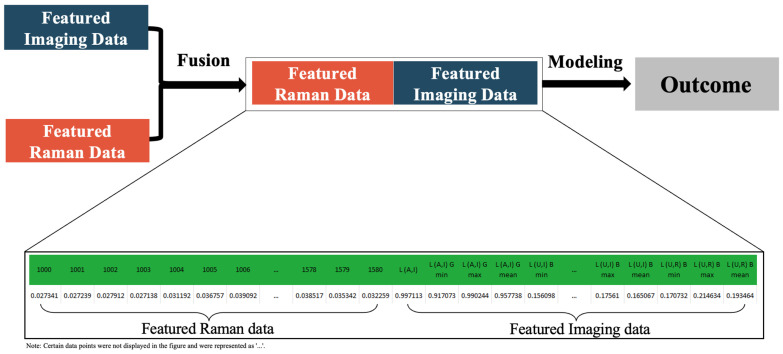
Schematic view of mid-level data fusion processing.

**Figure 4 foods-12-02322-f004:**
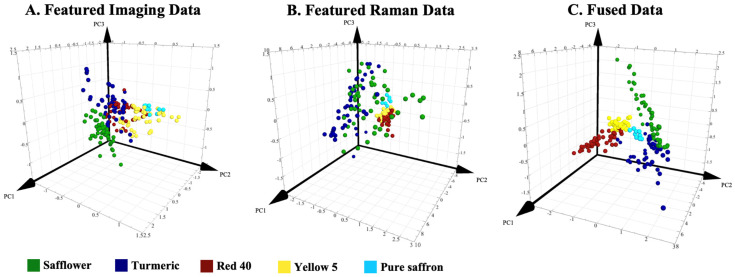
PLS-DA 3D-plots of (**A**) featured imaging data; (**B**) featured Raman data; (**C**) fused data.

**Figure 5 foods-12-02322-f005:**
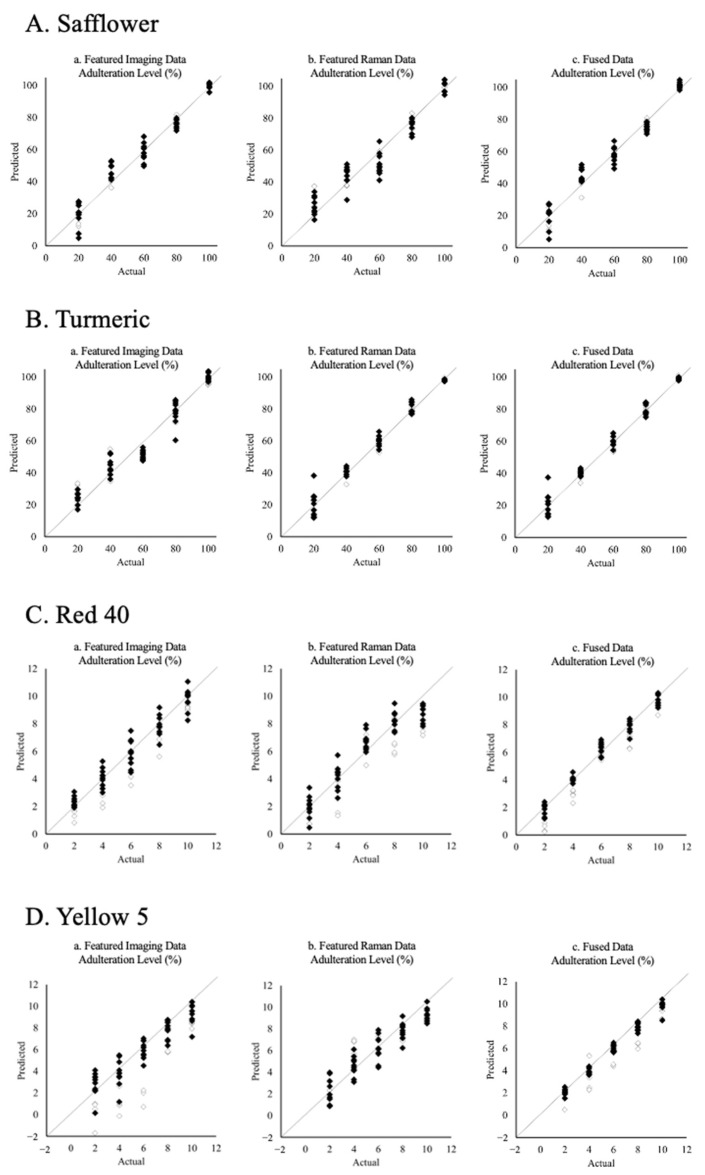
Predicted vs. actual PLS plots of spiked saffron samples using featured imaging, featured Raman, and fused data blocks for calibration (black diamond) and external validation (white diamond).

**Table 1 foods-12-02322-t001:** Misclassification table of spiked saffron samples based on (**A**) featured Imaging data, (**B**) featured Raman data, and (**C**) fused data as the training group.

	Members	Correct (%)	Safflower	Turmeric	Red 40	Yellow 5	Pure Saffron
	A. Featured Imaging data						
Safflower	50	100%	50	0	0	0	0
Turmeric	50	56%	2	28	20	0	0
Red 40	50	96%	0	0	48	2	0
Yellow 5	50	100%	0	0	0	50	0
Pure Saffron	10	0%	0	0	0	10	0
Total	210	83.81%	52	28	68	62	0
	B. Featured Raman Data						
Safflower	50	94%	47	3	0	0	0
Turmeric	50	94%	3	47	0	0	0
Red 40	50	80%	0	0	40	10	0
Yellow 5	50	72%	0	0	14	36	0
Pure Saffron	10	100%	0	0	0	0	10
Total	210	85.71%	50	50	54	46	10
	C. Fused Data						
Safflower	50	100%	50	0	0	0	0
Turmeric	50	100%	1	49	0	0	0
Red 40	50	98%	0	0	50	0	0
Yellow 5	50	100%	0	0	0	50	0
Pure Saffron	10	100%	0	0	0	0	10
Total	210	99.52%	51	49	50	50	10

Note: Highlighted numbers indicate accurately classified samples.

**Table 2 foods-12-02322-t002:** Misclassification table of the validation group based on the fused data block.

	Members	Correct (%)	Safflower	Turmeric	Red 40	Yellow 5	Pure Saffron
	Fused data (external validataion)						
Safflower	25	100%	25	0	0	0	0
Turmeric	25	100%	0	25	0	0	0
Red 40	25	96%	0	0	24	1	0
Yellow 5	25	100%	0	0	0	25	0
Pure Saffron	5	100%	0	0	0	0	5
Total	105	99.20%	25	25	24	26	5

Note: Highlighted numbers indicate accurately classified samples.

**Table 3 foods-12-02322-t003:** Statistics for each data block in external and cross-validations.

	Featured Imaging Data			Featured Raman Data			Fused Data		
	R^2^	RMSEP	RMSECV	R^2^	RMSEP	RMSECV	R^2^	RMSEP	RMSECV
Safflower	0.923	6.15	6.02	0.959	4.54	11.09	0.961	4.78	5.97
Turmeric	0.946	7.30	7.96	0.978	3.53	5.37	0.982	3.11	5.05
Red 40	0.923	1.28	0.82	0.892	1.65	1.37	0.978	1.03	0.97
Yellow 5	0.873	2.84	1.16	0.868	1.60	1.38	0.990	1.55	0.72

## Data Availability

The data presented in this study are available on request from the corresponding author.

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
