# Peer review of "Improving the Accuracy of Saffron Adulteration Classification and Quantification through Data Fusion of Thin-Layer Chromatography Imaging and Raman Spectral Analysis"

_foods, 2023, doi:10.3390/foods12122322_

Round 1
Reviewer 1 Report
Comments
This manuscript illustrated the data analysis of data fusion (between TLC images and associated Raman spectra). This technique would be used to classify and quantify saffron quality to determine its grade. This manuscript was clearly written and simple to comprehend. However, there are a few minor details that must be revised.
1. Author did not illustrate how to calculate X featured Raman . Is it related to xi* ? Please clarify
2. Author state the fusion dataset will be ??????= [????????? ?????,? ???????? ???????]. As the matrix dimension of X raman and X imaging (especially number of variables) is quite different. How can it combine into only single matrix as XFused. Please clarify.
3. In preprocessing, the author mentioned the "Par" scale type. This is just a SIMCA software-specific term. Author should provide more information about what "Par" means. in the manuscript for a wider audience.
4. From Figure 4, Is a PLSDA score plot? As the PLSDA model incorporates a class vector (supervised method) in the calculation. A risk of overfitting exists if the score plot is PLSDA. To be fair, the author should visualize a group of data using either unsupervised methods (e.g., PCA, HCA, etc.) or the PLSDA score of an external set.
5. Figure 5 depicts the calibration of the training dataset only ?? To demonstrate the model's efficacy for quantitative analysis of adulteration, the author should produce plots of both the training set and the external set (using distinct colors in the same plot).
Author Response
We greatly appreciate reviewers’ comments that help us to improve the manuscript. We have provided our responses and revised the manuscript accordingly. Thanks for your time.

Reviewer 2 Report
Dear Editor and Authors,
Regarding the manuscript ID: foods-2412740, entitled “Improving the Accuracy of Saffron Adulteration Classification and Quantification through Data Fusion of Thin-Layer Chromatography Imaging and Raman Spectral Analysis” my comments are the following.
The introduction section as well as the experimental are very well organized providing valuable information on the material studied. Although the same stands for the discussion section, I believe that further discussion and / or comparison with relative literature is missing. Also, the references that authors have sited are very few. Only 16 references, and none of them discussed or compared in the results section. I believe that further discussion and therefore extension of the reference and literature review is needed.
The manuscript it studies a very interesting subject. I suggest to make major revision on the results and discussion section.
Dear Editor and Authors,
Regarding the manuscript ID: foods-2412740, entitled “Improving the Accuracy of Saffron Adulteration Classification and Quantification through Data Fusion of Thin-Layer Chromatography Imaging and Raman Spectral Analysis” my comments are the following.
The introduction section as well as the experimental are very well organized providing valuable information on the material studied. Although the same stands for the discussion section, I believe that further discussion and / or comparison with relative literature is missing. Also, the references that authors have sited are very few. Only 16 references, and none of them discussed or compared in the results section. I believe that further discussion and therefore extension of the reference and literature review is needed.
I believe that the manuscript should be accepted for publication as it studies a very interesting subject. My suggestion is acceptance after major revision on the results and discussion section.
Author Response

(The authors gave the same response as above.)

Round 2
Reviewer 2 Report
Dear Editor and Authors,
Regarding the manuscript ID: foods-2412740, “Improving the Accuracy of Saffron Adulteration Classification and Quantification through Data Fusion of Thin-Layer Chromatography Imaging and Raman Spectral Analysis” I saw that the authors have taken my comments and remarks seriously into account and worked to improve the manuscript.
I suggest the acceptance of this study for publication.